

# Applications and limitations of current markerless motion capture methods for clinical gait biomechanics

Logan Wade[1,2], Laurie Needham[1,2], Polly McGuigan[1,2] and James Bilzon[1,2,3]

[1] Department for Health, University of Bath, Bath, United Kingdom
[2] Centre for Analysis of Motion, Entertainment Research and Applications, University of Bath, Bath, United Kingdom
[3] Centre for Sport Exercise and Osteoarthritis Research Versus Arthritis, University of Bath, Bath, United Kingdom

Corresponding author
Logan Wade, lw2175@bath.ac.uk

## ABSTRACT

**Background**. Markerless motion capture has the potential to perform movement analysis with reduced data collection and processing time compared to marker-based methods. This technology is now starting to be applied for clinical and rehabilitation applications and therefore it is crucial that users of these systems understand both their potential and limitations. This literature review aims to provide a comprehensive overview of the current state of markerless motion capture for both single camera and multi-camera systems. Additionally, this review explores how practical applications of markerless technology are being used in clinical and rehabilitation settings, and examines the future challenges and directions markerless research must explore to facilitate full integration of this technology within clinical biomechanics.

**Methodology**. A scoping review is needed to examine this emerging broad body of literature and determine where gaps in knowledge exist, this is key to developing motion capture methods that are cost effective and practically relevant to clinicians, coaches and researchers around the world. Literature searches were performed to examine studies that report accuracy of markerless motion capture methods, explore current practical applications of markerless motion capture methods in clinical biomechanics and identify gaps in our knowledge that are relevant to future developments in this area.

**Results**. Markerless methods increase motion capture data versatility, enabling datasets to be re-analyzed using updated pose estimation algorithms and may even provide clinicians with the capability to collect data while patients are wearing normal clothing. While markerless temporospatial measures generally appear to be equivalent to marker-based motion capture, joint center locations and joint angles are not yet sufficiently accurate for clinical applications. Pose estimation algorithms are approaching similar error rates of marker-based motion capture, however, without comparison to a gold standard, such as bi-planar videoradiography, the true accuracy of markerless systems remains unknown.

**Conclusions**. Current open-source pose estimation algorithms were never designed for biomechanical applications, therefore, datasets on which they have been trained are inconsistently and inaccurately labelled. Improvements to labelling of open-source training data, as well as assessment of markerless accuracy against gold standard methods will be vital next steps in the development of this technology.

## INTRODUCTION

Movement analysis seeks to understand the cause of altered movement patterns, assisting with prevention, identification and rehabilitation of a wide array of diseases, disabilities and injuries (*Astephen et al., 2008*; *Franklyn-Miller et al., 2017*; *Hausdorff et al., 2000*; *Heesen et al., 2008*; *King et al., 2018*; *Pavão et al., 2013*; *Salarian et al., 2004*; *Sawacha et al., 2012*; *Vergara et al., 2012*). In modern medicine, early identification now plays a major role in combating disease progression, facilitating interventions using precise measurements of small changes in movement characteristics (*Buckley et al., 2019*; *Noyes & Weinstock-Guttman, 2013*; *Rudwaleit, Khan & Sieper, 2005*; *Swash, 1998*). Movement analysis may also assist with injury prevention in athletes (*Paterno et al., 2010*), improve rehabilitation treatment and adherence (*Knippenberg et al., 2017*), and may inform surgical intervention methods to optimize outcomes and reduce additional surgeries and healthcare costs (*Arnold & Delp, 2005*; *Jalalian, Gibson & Tay, 2013*; *Lofterød et al., 2007*; *Wren et al., 2009*).

Traditional movement analysis commonly relies on patient self-reports, along with practitioner observations and visually assessed rating scales to diagnose, monitor and treat musculoskeletal diseases (*Berg et al., 1992*; *Jenkinson et al., 1994*; *Zochling, 2011*). Unfortunately, these measures are often subjective and prone to error, as they are based on each individual's interpretation (*Muro-de-la Herran, Garcia-Zapirain & Mendez-Zorrilla, 2014*). Wearable devices such as inertial measurement units (IMU) can provide clinicians with motion capture data that is quantitative, reliable and relatively easy to collect. There have been numerous reviews assessing the pros and cons of IMU devices (*Baker, 2006*; *Buckley et al., 2019*; *Chen et al., 2016*; *Muro-de-la Herran, Garcia-Zapirain & Mendez-Zorrilla, 2014*; *Tao et al., 2012*) and therefore while still potentially an area of interest, these devices will not be a focus of this review. Alternatively, video-based motion capture records and processes video images to identify limb location and orientation, enabling calculation of output variables such as temporospatial measures and joint angles. Describing the position and orientation or 'pose' of body segments in three-dimensions (3D) requires calculation of the limbs' translation (sagittal, frontal and transverse position, Fig. 1) and rotation (flexion/extension, abduction/adduction and rotation about the longitudinal axis, Fig. 1). These three translational and three rotational descriptions of a segment are commonly referred to as six degrees of freedom (DoF). The current gold standard for non-invasive video-based motion capture, is bi-planar videoradiography, which uses multiple X-ray views to capture video of bone movement (*Kessler et al., 2019*; *Miranda et al., 2011*). Software is used to outline the bones and recreate their three-dimensional structure (*Kessler et al., 2019*), enabling 3D joint center locations and angles to be extracted with high precision. However, even this method has joint center translational errors of 0.3 mm and rotational errors of 0.44° (*Miranda et al., 2011*). Additionally, high costs,

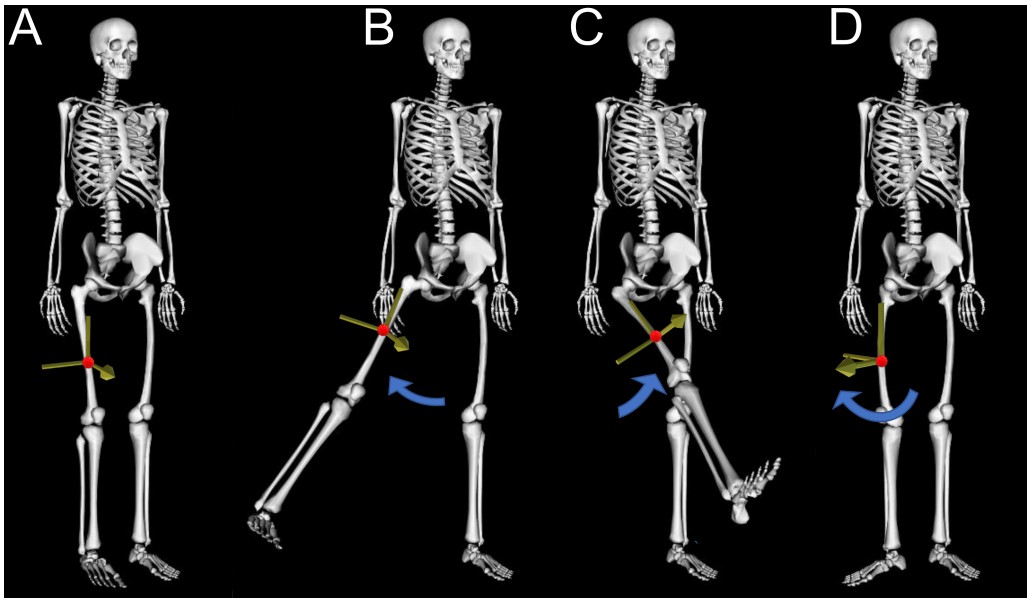

**Figure 1 Six degrees of freedom.** This figure demonstrates the six degrees of freedom needed to describe position and orientation (pose) of the human body, with the red dot indicating the location (translation) of the segment center of mass and blue arrows indicating rotation in three planes. (A) The reference standing posture, (B) thigh segment adduction/abduction, (C) thigh segment flexion/extension, (D) thigh segment rotation about the longitudinal axis.

small capture volume (single joint) and exposure to radiation make clinical or sporting applications impractical.

Due to bi-planar videoradiography limitations, the de facto video-based motion capture method is marker-based motion capture, which identifies human poses using near-infrared cameras and reflective markers placed on the skin (Fig. 2). Marker locations can be detected with sub-millimeter accuracy (*Buckley et al., 2019*; *Topley & Richards, 2020*) and are used to identify location and orientation of body segments for calculation of joint positions and angles. However, marker-based motion capture has significant drawbacks, requiring a controlled environment (*Buckley et al., 2019*; *Chen et al., 2016*) that may alter participants movements, due to their awareness of being under observation (*Robles-García et al., 2015*). Marker-based systems are cheaper to acquire and run compared to biplanar videoradiography, but are generally still too expensive for many clinical applications, as highly trained personnel are required to operate them (*Simon, 2004*). Marker-based motion capture also suffers from human error when placing markers on the participant (*Gorton, Hebert & Gannotti, 2009*), and marker placement is very time intensive which can be a significant barrier in clinical or sporting environments, particularly with specific population groups (*Whittle, 1996*).

While highly popular, marker-based motion capture is not a gold standard, despite often being treated as such. Comparisons of marker-based motion capture against biplanar videoradiography reveal joint center position errors across the body as high as 30 mm, with averages between 9 and 19 mm, and joint rotation errors across the body as high

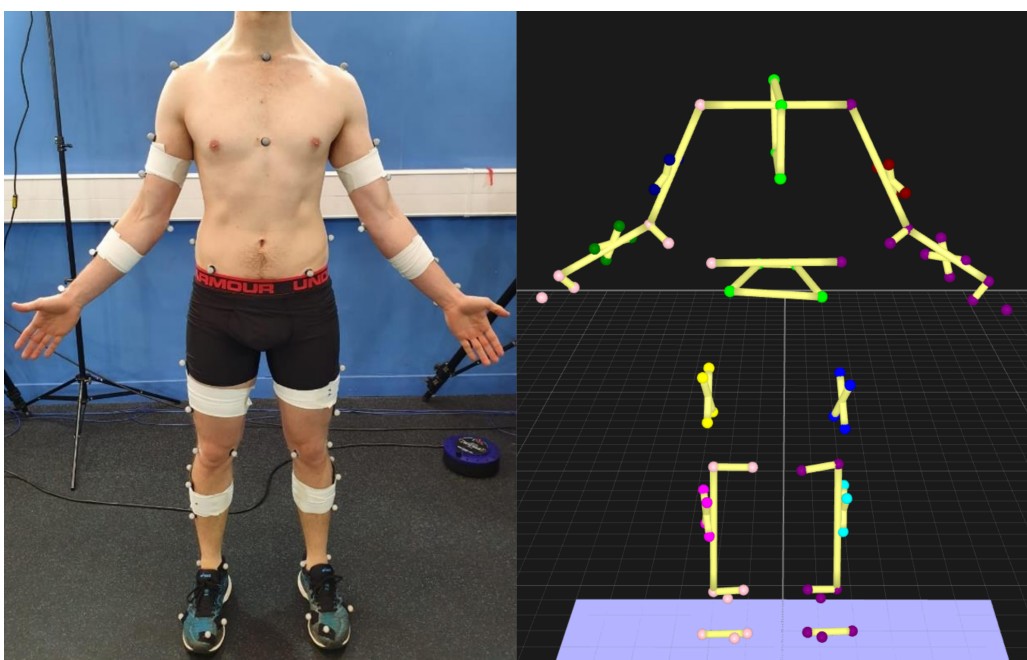

**Figure 2** **Optoelectronic motion capture markers.** Left—markers placed on the participant. Right—view of the markers in 3D space.

as 14°, with averages between 2.2 and 5.5° (*Miranda et al., 2013*). For all motion capture methods, rotation about the longitudinal axis (Fig. 1D) produces the greatest errors of all rotational planes (*Kessler et al., 2019*; *Miranda et al., 2013*) as measurement devices placed on the skin (*i.e.,* markers) are much closer to the axis of rotation, with hip internal-external rotational errors possibly as high as 21.8° (*Fiorentino et al., 2017*).

Marker-based errors are partially due to an assumption that markers on the skin represent position of the bone. However, this assumption leads to soft tissue artefact errors as muscle, fat and skin beneath markers cause them to move independently from bone (*Camomilla, Bonci & Cappozzo, 2017*; *Cappozzo et al., 1996*; *Peters et al., 2010*; *Reinschmidt et al., 1997*). Compared to bi-planar videoradiography, errors for markers placed over shank soft tissue were 5–7 mm, while markers placed over bony landmarks on the foot were 3–5 mm (*Kessler et al., 2019*). Soft tissue errors for hip joint range of motion may be on average between 4 and 8° during walking, stair descent and rising from a chair (*D'Isidoro, Brockmann & Ferguson, 2020*). Procedures such as filtering the marker data can help to reduce some of this soft tissue error (*Camomilla, Bonci & Cappozzo, 2017*; *Peters et al., 2010*). However, without invasively attaching markers to bone this error cannot be eliminated (*Benoit et al., 2006*) and therefore, soft tissue artefact will continue to limit the accuracy of marker-based methods.

There is a need for motion capture methods that are less time intensive, do not require specialist personnel, and are less impacted by errors associated with marker-based methods (*e.g.,* soft tissue artefact). Markerless motion capture uses standard video to record movement without markers, often leveraging deep learning-based software to identify

body segment positions and orientations (pose). However, this technology has been slow to transfer to biomechanics, likely due to the requirement of advanced coding skills and in-depth computer science knowledge. As such, researchers, clinicians and coaches using this technology need to be informed of the benefits and limitations of these methods. Currently, there are no reviews targeted at applications of markerless motion capture for clinical biomechanics and sports medicine, which we aim to resolve within this review. This scoping review is intended to inform clinical biomechanical researchers, clinicians and coaches of current markerless motion capture performance, explore how this technology can be used in real world applications and discuss future directions and limitations that need to be overcome for markerless systems to become viable for clinical, rehabilitation and sporting applications.

## SURVEY METHODOLOGY

A scoping review is needed to examine this emerging broad body of literature and determine where gaps in knowledge exist (*Munn et al., 2018*). This examination is key to developing motion capture methods that are cost effective and practically relevant to clinicians, coaches and researchers around the world. Literature searches were performed to target studies that report accuracy of markerless motion capture methods compared to marker-based motion capture or manually labelled methods. Literature searches were then performed to target current practical applications of markerless motion capture methods in clinical biomechanics. Finally, examination of markerless motion capture literature was performed to determine what gaps in the knowledge exist and discuss future directions and limitations of this developing technology. Literature was obtained using Google Scholar and Scopus, which were surveyed using different combinations of the keywords 'markerless', 'motion capture', 'pose estimation', 'gait analysis', 'clinical biomechanics', 'accuracy', '2D' and '3D', without limits on publication date. Literature was also obtained from reference lists of identified articles.

## MARKERLESS MOTION CAPTURE

Markerless motion capture uses standard video and often relies on deep learning-based software (pose estimation algorithms) to describe human posture for each individual image within the video, or videos for multiple cameras (Fig. 3). Because pose estimation algorithms are not dependent on markers attached to the skin, soft tissue artefact errors may be reduced compared to marker-based methods, although this is yet to be examined experimentally. Pose estimation algorithms can be applied to new or old videos, provided sufficient image resolution, and while marker-based methods are limited by the marker-set used during data collection, old markerless video data could be reprocessed with new pose estimation algorithms to improve accuracy or extract more in-depth measures. Accurate application of this technology could therefore facilitate streamlined monitoring of changes in disease progression (*Kidziński et al., 2020*), rehabilitation (*Cronin et al., 2019*; *Natarajan et al., 2017*), athletic training and competition (*Evans et al., 2018*), and injury prevention (*Zhang, Yan & Li, 2018*).
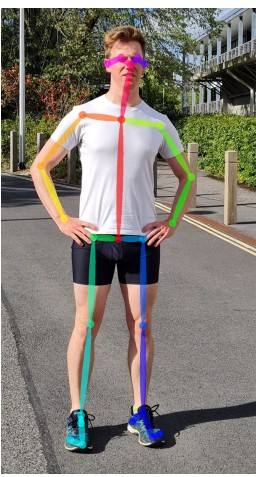

**Figure 3** Twenty-five keypoints detected using the OpenPose pose estimation algorithm (*Cao et al., 2018*) applied to a single image.

## Hardware

The two main types of camera hardware employ either depth cameras or standard video cameras and may be used in single or multi-camera systems. Depth cameras, such as the Microsoft Azure Kinect, record standard video and additionally also record the distance between each pixel and the camera (depth). While depth cameras are relatively cheap and accessible, research has demonstrated large differences compared to marker-based methods (*Dolatabadi, Taati & Mihailidis, 2016*; *Mentiplay et al., 2015*; *Natarajan et al., 2017*; *Otte et al., 2016*; *Pantzar-Castilla et al., 2018*; *Rodrigues et al., 2019*; *Tanaka et al., 2018*). Additionally, depth cameras have limitations on capture rate, capture volume and data collection may require controlled lighting conditions (*Clark et al., 2019*; *Sarbolandi, Lefloch & Kolb, 2015*). There have been several in-depth reviews of these systems (*Clark et al., 2019*; *Garcia-Agundez et al., 2019*; *Knippenberg et al., 2017*; *Mousavi Hondori & Khademi, 2014*; *Webster & Celik, 2014*) and while depth cameras are still an active area of research, this review will focus on single and multi-camera markerless systems that use standard video cameras, as these systems are relatively new and have recently started to be employed for clinical, rehabilitation and injury prevention applications.

Markerless motion capture using standard video hardware does have some limitations similar to marker-based systems, as the capture volume is still limited by the number of cameras and high-speed cameras require much brighter lighting. However, compared to marker-based systems that rely on infrared cameras, markerless motion capture is not limited by sunlight or multiple systems running simultaneously. Zoom lenses or high-resolution video can enable data collection from long distances and is currently being used during sporting competitions such as tennis (Hawk-Eye) and baseball (Kinatrax) to track the ball and players. Low-cost systems could employ webcams or smartphones to record video data, facilitating motion capture by clinicians and coaches in real world applications. Higher end multi-camera systems that record synchronized video at high
frame rates may be used for collection of high precision data, akin to current marker-based motion capture laboratories. However, extracting meaningful information (joint centers) from recorded images using software is a very difficult task to perform with high accuracy.

## Software

Once video data is collected, software in the form of pose estimation algorithms are employed to detect and extract joint center locations. Pose estimation algorithms typically use machine learning techniques that allow them to recognize patterns associated with anatomical landmarks. These algorithms are 'trained' using large scale datasets that provide many examples of the points of interest. However, to a computer, video data is comprised of pixels that are essentially a grid of numbers, with each number in the grid describing color and brightness in a given video frame, which makes identifying keypoints a very challenging task. Training a pose estimation algorithm generally requires the creation of a dataset containing thousands of manually labelled keypoints (Fig. 4) (*Chen, Tian & He, 2020*; *Ionescu et al., 2014*; *Lin et al., 2014*; *Sigal, Balan & Black, 2010*). Deep learning-based pose estimation algorithms perform mathematical calculations on each image in the training data, using a layered network (Convolutional Neural Network) that may be many layers deep (*Mathis et al., 2020b*), where the output of one layer becomes the input of the next layer (Fig. 4). In doing this, a pose estimation algorithm learns to identify keypoints (*e.g.*, joint centers) as patterns of pixel color, gradient and texture from the training data. Distance between the manually labelled and estimated keypoint locations are then examined by an optimization method, which updates filters within each layer of the pose estimation algorithm to reduce the distance between keypoints (Fig. 4). This process is repeated using the entire training dataset until improvements between each iteration become negligible (Fig. 4). The pose estimation algorithm is then tested on new images and compared to manually labelled data or marker-based joint center locations to determine how well it performs on images it has never seen. As such, deep learning-based pose estimation will only ever be as good as the training data used.

Two pose estimation algorithms that have become very popular for biomechanical applications are OpenPose (*Cao et al., 2018*) and DeepLabCut (*Insafutdinov et al., 2016*; *Mathis et al., 2018*). OpenPose is a powerful pose estimation algorithm that can track multiple people in an image and is very easy to use. DeepLabCut enables users to retrain/refine a pre-trained pose estimation algorithm by providing the algorithm with a subset of manually labelled images that are specific to the desired task (∼200 images) (*Mathis et al., 2018*), which can be especially useful for uncommon movements (*e.g.*, clinical gait or sporting movements). For an in-depth review of pose estimation algorithm designs, readers are directed to numerous alternative reviews (*Chen, Tian & He, 2020*; *Colyer et al., 2018*; *Dang et al., 2019*; *Mathis et al., 2020a*; *Sarafianos et al., 2016*; *Voulodimos et al., 2018*).

While marker-based motion capture relies heavily on hardware (markers physically placed on the skin) to extract segment poses (location and orientation), markerless motion capture relies on software to process the complicated image data obtained by standard video hardware (as explained above). Unfortunately, most of the current pose estimation algorithms have been trained to only extract two points on each segment (proximal

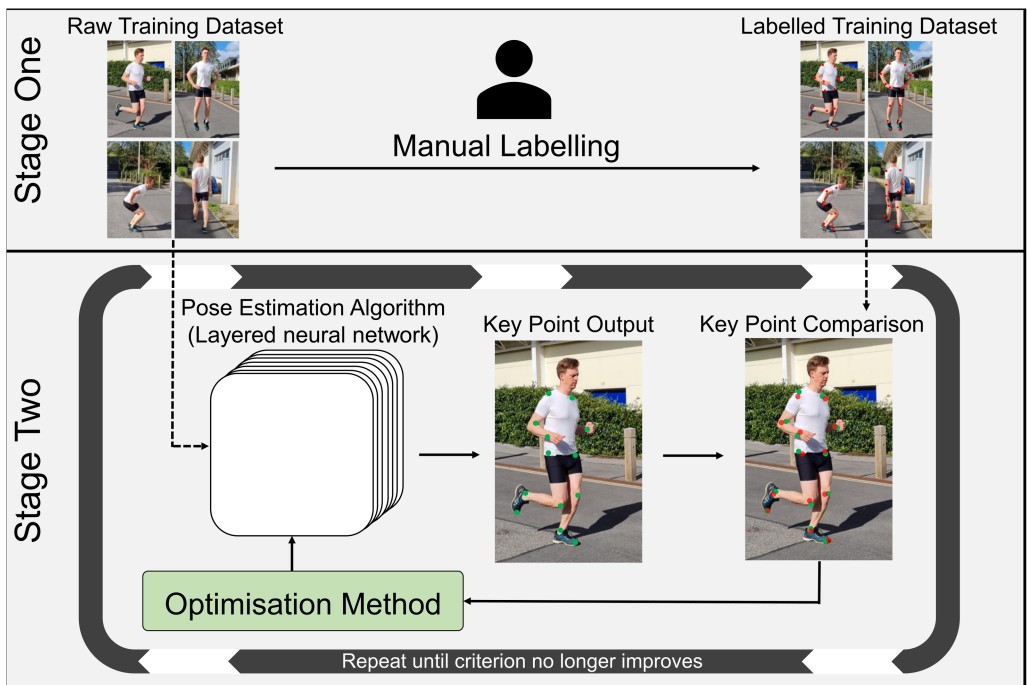

**Figure 4** **Pose estimation algorithm training workflow.** Stage One: Creation of a manually labelled training dataset. Stage Two: Using the unlabeled images from stage one, the pose estimation algorithm estimates the desired keypoint locations (joint centers). Estimated keypoint locations are then compared to the manually labelled training data from stage one, to determine the distance between the estimated keypoint and the manually labelled keypoint. The optimization method then adjusts filters within the layers of the algorithm to try to reduce this distance and new estimated keypoints are calculated. This process is repeated until improvements to the pose estimation algorithm are negligible.

and distal joint center locations), whilst three keypoints are required to calculate 6DoF (*e.g.*, proximal and distal end of a segment, and a third point placed somewhere else on the segment). Two keypoints can provide information about the sagittal and coronal planes (Figs. 1B and 1C), while the third keypoint is needed to determine rotation about the segment's longitudinal axis (Fig. 1D). Thus, markerless methods that only identify joint center locations are limited to 5DoF, which only enables examination of 2D planar joint angles. This may be overcome to some degree by combining 5DoF methods with musculoskeletal modelling to constrain the movement and estimate movement in 6DoF (*Chen & Ramanan, 2017*; *Gu et al., 2018*), however, manually relabeling training data with an additional third keypoint location on each segment will likely produce improved results with less processing of the data (*Needham et al., 2021b*).

Markerless motion capture has been slow in transferring to biomechanics, primarily due to inaccuracy of detecting joint center locations (*Harsted et al., 2019*) and requiring knowledge of computer vision and advanced programming skills. In this review, we have classified markerless motion capture into two broad categories: monocular markerless motion capture which uses a single camera, and multi-camera markerless motion capture which obtains video data from two or more synchronized cameras. Despite its previously

2D Pose Estimation from Monocular Motion Capture

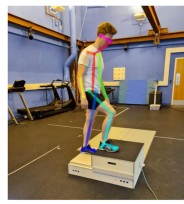 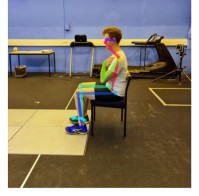

3D Pose Estimation from Multi-camera Motion Capture

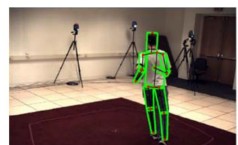 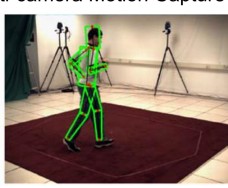

3D Pose Estimation from Monocular Motion Capture

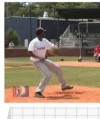 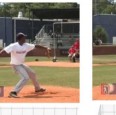 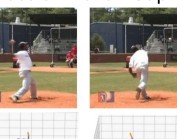

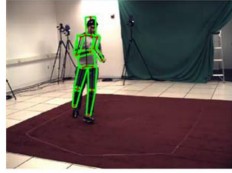 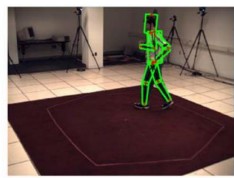

**Figure 5  2D and 3D pose estimation.** Markerless motion capture examples: 2D pose estimation from monocular motion capture (2D keypoints detected using OpenPose *Cao et al. (2018)*), 3D pose estimation from monocular motion capture (adapted from *Cheng et al. (2020)* with license from the Association for the Advancement of Artificial Intelligence, Copyright ©2020) and 3D pose estimation from multi-camera motion capture (adapted from *Sigal, Balan & Black (2010)* with permission from Springer Nature).

outlined faults, marker-based motion capture has generally been used as the reference method when assessing accuracy of markerless motion capture, and this should be kept in mind when comparing results between systems.

## PERFORMANCE OF CURRENT MARKERLESS APPLICATIONS

### Monocular markerless motion capture

2D monocular markerless motion capture obtains joint center locations from a single image or video using 2D pose estimation algorithms (Fig. 5), making it cost and space efficient. However, self-occlusion errors are a major issue, often causing joint center locations to be missing for one or more frames and contribute to instances where the opposite limb is incorrectly detected (*e.g.*, right knee labelled as the left knee) (*Serrancolí et al., 2020*; *Stenum, Rossi & Roemmich, 2021*). Similar to marker-based methods, obtaining biomechanically relevant 2D planar joint angles requires an assumption that the camera is perfectly aligned with frontal or sagittal plane movements (*Stenum, Rossi & Roemmich, 2021*). If correctly aligned with the plane of action (1DoF), the pose estimation method detects the translational joint center coordinates in the horizontal and vertical axes (2DoF), which are then combined with coordinates of neighboring joints to calculate 2D rotational segment and joint angles (3DoF).

Three studies have examined 2D monocular applications (25–60 Hz) of DeepLabCut against manual labelling or marker-based methods for the leg closest to the camera (sagittal view), in underwater running (*Cronin et al., 2019*), countermovement jumping (*Drazan et al., 2021*) and walking in stroke survivors (*Moro et al., 2020*). Markerless joint center differences were 10–20 mm greater than marker-based motion capture, but no significant
differences were found between methods for temporospatial and joint angle outcome measures during walking and underwater running, and therefore this method may be a suitable alternative to 2D marker-based motion capture (*Cronin et al., 2019*; *Moro et al., 2020*). Strong correlations were found for joint angles during countermovement jumping compared to marker-based methods, however this study had to perform a knee and hip correction based on marker-based results (5.6°). Therefore, it is unknown if these systematic offsets would be applicable for future applications.

While not strictly monocular, *Serrancolí et al. (2020)* and *Stenum, Rossi & Roemmich (2021)* used two video cameras (25–60 Hz), placed on either side of a person, to extract information of the side closest to each camera and negate occlusion errors during walking over-ground or cycling on an ergometer. During walking, temporal differences were on average within 1 frame and spatial differences were less than one cm, although maximum differences were as high as 20 cm (*Stenum, Rossi & Roemmich, 2021*). For both studies, lower limb joint angle differences were 3–11 degrees greater than marker-based methods and thus were too large to detect small changes needed for real world applications. Both studies also required additional manual input to fix incorrectly detected joints (*e.g.*, right knee labelled as the left knee) (*Serrancolí et al., 2020*; *Stenum, Rossi & Roemmich, 2021*). Therefore, some 2D monocular methods may obtain temporospatial (DeepLabCut and OpenPose) and planar 2D joint angles (DeepLabCut) with accuracy similar to marker-based motion capture (*Miranda et al., 2013*), but this has only been examined for the side of the body closest to the camera. 2D motion capture will likely have the most value in general clinical or rehabilitation environments, where data collection can be tightly controlled to reduce occlusion issues, and decreasing data collection and processing time is paramount.

Obtaining 3D joint center locations from monocular markerless motion capture (Fig. 5) seeks to estimate joint locations in 3D using a single standard camera that only records 2D images (*Mehta et al., 2017*). However, because the participant may move in any direction (plane), entire limbs may be occluded for significant periods. Additionally, depth must be estimated from 2D video data to determine which joints are closer to the camera (*Chen, Tian & He, 2020*). Obscured 3D joint locations may be estimated using past or future un-occluded frames, or from the position of un-occluded neighboring joints in the current frame (*Cheng et al., 2020*; *Cheng et al., 2019*; *Khan, Salahuddin & Javidnia, 2020*; *Mehta et al., 2017*; *Moon, Chang & Lee, 2019*; *Yang et al., 2018*). Alternatively, 2D monocular methods may be combined with musculoskeletal modelling (*Chen & Ramanan, 2017*; *Gu et al., 2018*) or estimation of forces (*Rempe et al., 2020*), to restrict the limb position in 3D and assist with unnatural leaning angles towards or away from the camera (*Rempe et al., 2020*). Multi-camera marker-based motion capture can be used to train monocular pose estimation methods to make an educated guess about where a joint is in 3D, however due to a fundamental lack of data (*i.e.,* there is no information about occluded joints from single camera images), this will only ever be an estimate. Finally, as mentioned earlier, current pose estimation methods generally only detect two points on a segment (proximal and distal joint center locations) (*Cao et al., 2018*; *Mathis et al., 2018*), which can only measure 5DoF. Thus, manually relabeling training data to detect a third point on each segment could improve estimation of 6DoF.

3D monocular joint center location differences compared to reference methods are generally 40–60 mm (*Chen, Tian & He, 2020*), with some algorithms producing 30–40 mm differences when specifically trained to overcome occlusion issues (*Cheng et al., 2020*; *Cheng et al., 2019*). 3D monocular ankle joint angle differences during walking are between −10° and 10° for normal walking with maximal differences of 30° compared to marker-based methods. Two studies have examined temporospatial measures (step length, walking speed and cadence) using 2D monocular methods combined with projection mapping (*Shin et al., 2021*) or a 3D musculoskeletal model (*Azhand et al., 2021*), finding strong correlations when compared to the GAITRite pressure walkway (*Azhand et al., 2021*; *Shin et al., 2021*). Therefore, while temporospatial measures may have sufficient accuracy for real world applications, significant improvements to identification of joint center location and angle are needed. Applications of this method will likely also require the user to minimize instances where limbs are fully occluded (*e.g.*, setting up the camera in the frontal plane) (*Shin et al., 2021*).

## Multi-camera markerless motion capture

Multi-camera markerless motion capture is a progression of 2D monocular methods that minimizes joint occlusion errors by employing multiple cameras (Fig. 5). This method combines 2D pose estimation with an additional multi-camera reconstruction step to estimate 3D joint center locations (*Nakano et al., 2020*; *Needham et al., 2021a*; *Slembrouck et al., 2020*). Compared to monocular systems, multi-camera systems are more costly due to additional hardware and require more space, thus this method generally seeks to replicate the results obtained from current high-end marker-based systems (*e.g.*, Qualisys/Vicon).

Several studies have examined multi-camera markerless systems using the OpenPose pose estimation algorithm (30–120 Hz), reporting average joint center location differences between 10 and 50 mm (*Nakano et al., 2020*; *Slembrouck et al., 2020*; *Zago et al., 2020*) and temporospatial differences of 15 mm compared to marker-based methods (*Zago et al., 2020*). Slower movements had better results, with average walking joint center differences compared to marker-based methods of 10–30 mm, while faster jumping and throwing movements were 20–40 mm (*Nakano et al., 2020*), which may be exacerbated with slow video frame rates (*Slembrouck et al., 2020*; *Zago et al., 2020*). Manual adjustments were required when OpenPose incorrectly detected joints (*e.g.*, detects left knee as the right knee) for one study (*Nakano et al., 2020*). *Needham et al. (2021b)* performed a recent comparison of OpenPose (*Cao et al., 2018*), DeepLabCut (*Mathis et al., 2018*) and a third pose estimation algorithm (AlphaPose (*Fang et al., 2017*)) using 9 video cameras and 15 marker-based cameras both collecting at 200 Hz. Compared to marker-based methods, 3D lower limb joint center differences were smallest for OpenPose and AlphaPose at 16–34 mm during walking, 23–48 mm during running and 14–36 mm during jumping. It should be noted that they did not retrain models using DeepLabCut and instead used the DeepLabCut standard human pose estimation algorithm (*Mathis et al., 2018*). While these results are now approaching error rates of marker-based motion capture identified by Miranda and colleagues (*2013*), Needham and colleagues demonstrated that there were systematic differences for all markerless methods, with the largest systematic differences

occurring at the hip. Their paper suggested this is likely the product of poorly labelled open access datasets, with the hip joint being the worst, as this joint is very difficult to identify correctly without physical palpation and therefore, these inaccuracies may limit detection of reliable joint center locations.

While previous studies have used open-source pose estimation algorithms and therefore may be considered as standalone experimental setups, commercial systems have been developed. Joint angles were compared between an eight camera (50 Hz) Captury markerless system (Captury) and a 16 camera marker-based system, although Captury identifies the silhouette of a person instead of using deep learning to extract joint center locations (*Harsted et al., 2019*). The authors stated that planar joint angles could not be considered interchangeable between motion capture systems, with lower limb joint angle differences of 4–20°. Alternatively, another study employed SIMI Reality Motion Systems to record multiple movements using eight cameras (100 Hz). Images were processed using either Simi Motion software, which detects markers placed on the skin, or Simi Shape 3D software, which is a markerless software that uses silhouette-based tracking similar to Captury (*Becker, 2016*). Standard deviations of lower limb joint angles were between 3 and 10 degrees with the markerless method compared to the marker-based method, and correlations for hip and ankle frontal and rotation planes were poor (0.26–0.51), indicating high variability of this system. Most recently, Theia3D markerless software (Theia Markerless Inc.) which uses a proprietary pose estimation algorithm was compared between an 8 camera markerless system (85 Hz) and a seven camera marker-based system (85 Hz) (*Kanko et al., 2021b*; *Kanko et al., 2021c*). They reported no bias or statistical difference for walking spatial measures (*e.g.*, step length, step width, velocity) and a small difference in temporal measures (*e.g.*, swing time and double support time) (*Kanko et al., 2021c*). A follow-on study using the same data found average differences of 22–36 mm for joint centers and 2.6–11 degrees for flexion/extension and abduction/adduction, although rotation about the longitudinal axis differences were 6.9–13.2 degrees compared to marker-based methods (*Kanko et al., 2021b*). Importantly, the lower ranges of these translational and rotational differences are within error rates identified by previous research (*Fiorentino et al., 2017*; *Kessler et al., 2019*; *Miranda et al., 2013*). These strong results appear to be due to Theia3D having labelled their own biomechanically applicable data set which identifies 51 keypoints on the body (*Kanko et al., 2021b*; *Kanko et al., 2021c*), compared to OpenPose which only identifies 25 points (*Cao et al., 2018*). However, Theia3D software is somewhat of a black box, as it is unknown exactly which keypoints are being used. Now that some markerless systems are approaching the accuracy of marker-based methods, which have known errors discussed previously, future examination of markerless accuracy will require comparison to a gold standard method such as bi-planar videoradiography (*Miranda et al., 2013*).

## PRACTICAL APPLICATIONS

While markerless systems may still be considered in their infancy, there have been several studies that demonstrate markerless potential for clinical applications. DeepLabCut was

used to extract walking sagittal 2D joint angles in stroke survivors, showing significant differences between the affected and unaffected side (*Moro et al., 2020*). *Cunningham et al. (2019)* examined 2D monocular segment angles of a multi-segmented trunk and head in young children with cerebral palsy, enabling automation of clinical tests to examine spine and head posture. *Baldewijns et al. (2016)* measured walking speed recorded unobtrusively in patient's homes using a webcam, demonstrating how markerless methods could provide continuous monitoring of patients as they go about their daily lives. *Martinez et al. (2018)* used a 2D monocular markerless system with OpenPose to examine walking cadence and automate calculation of an anomaly score for Parkinson's disease patients, providing clinicians with an unbiased general overview of patient disease progression. Finally, *Shin et al. (2021)* retrospectively analyzed monocular frontal videos of Parkinson's patients for temporospatial outcome measures (step length, walking velocity and turning time). They demonstrated high correlations between subjective clinical gait tests and were able to detect minor gait disturbances unnoticed by the clinician.

In one significant clinical example, *Kidziński et al. (2020)* analyzed 2D outcomes of cerebral palsy gait collected from a single camera (30 Hz) between 1994 and 2015 (~1,800 videos). OpenPose derived 2D joint centers were used as the input for a secondary deep learning-based neural network that predicted parameters of clinical relevance, such as walking speed, cadence and knee flexion angle. However, direct comparisons to marker-based methods could not be performed due to data collection methods and therefore, new test data collected simultaneously with marker-based motion capture is needed to examine the accuracy of their system. Nevertheless, this study compiled outcome measures into a gait report that was automatically generated for the clinician, providing strong rationale for the future of clinical biomechanics and its ability to analyze gait in a cost and time efficient manner. Furthermore, the applications by *Kidziński et al. (2020)* and *Shin et al. (2021)* highlight the value of markerless motion capture to extract new information from old datasets. Without the need to place markers on participants or manually process results, quantitatively tracking patients throughout disease progression and rehabilitation becomes a much more viable option.

While some markerless systems may be approaching the accuracy of marker-based methods, some applications may not need highly accurate data and instead, numerous trials (*e.g.*, numerous walking strides) could be averaged to obtain reliable average results (*Pantzar-Castilla et al., 2018*). Unfortunately, this approach may be unable to detect small changes over time and it is not always possible to collect many trials in a clinical, rehabilitation or sport setting. Alternatively, using markerless motion capture as a motivational tool to perform rehabilitation exercises does not require highly accurate results. Markerless motion capture can be used to control a game or move around a virtual environment, which can increase adherence and motivation to perform repetitive or potentially painful rehabilitation exercises (*Knippenberg et al., 2017*; *Vonstad et al., 2020*). This could lead to improved rehabilitation methods, as interaction with virtual environments has also been shown to reduce pain felt by patients (*Gupta, Scott & Dukewich, 2017*; *Scapin et al., 2018*). While this application has been used with depth cameras (*e.g.*,

Microsoft Kinect) (*Chanpimol et al., 2017*; *Knippenberg et al., 2017*), current applications using standard cameras and pose estimation algorithms are limited (*Clark et al., 2019*).

## FUTURE CHALLENGES AND APPLICATIONS

### Clothing

Currently, markerless systems are assessed while participants wear tight fitting clothing, as marker-based motion capture cannot be used with normal/baggy clothing. However, normal clothing is often loose fitting and may change shape during movement, which may or may not impact a pose estimation algorithms ability to accurately extract joint center locations (*Sarafianos et al., 2016*). If markerless systems are resistant to this issue, it could greatly improve efficiency and ease of data collection in clinical and real-world applications. Using eight cameras (60 Hz) with Theia3D's pose estimation algorithm, inter-trial and inter-session joint angle variability during walking was examined compared to previously reported marker-based results (*Kanko et al., 2021a*). Participants wore their own clothing which generally consisted of shoes, long trousers, shirt and sweater. Markerless inter-trial joint angle variability was on average 2.5°, compared to 1.0° from marker-based methods (*Kanko et al., 2021a*; *Schwartz, Trost & Wervey, 2004*), while markerless inter-session variability was on average 2.8° compared to 3.1° for marker-based methods (*Kanko et al., 2021a*; *Schwartz, Trost & Wervey, 2004*). Therefore, markerless joint angle variability within the same day and across multiple days (intra-session and inter-session), may be similar to marker-based data collected on multiple days (inter-session). Testing across multiple days or changes of clothing had no impact on the overall variability of the markerless system. However, the higher inter-trial variability suggests that markerless methods do produce greater errors during the same session. Unfortunately, because they did not examine marker-based walking variability of their participants, it is unknown if variability from previous marker-based studies was identical to that exhibited by the participants included within this study. Importantly, markerless data collection was able to be completed in 5–10 min, demonstrating the benefits of this system for applications where time is limited (*Kanko et al., 2021a*). Based on these results, markerless systems could one day collect data on patients at home during daily life, without the need of an operator or tight-fitting clothing. Such systems could also be set up in common areas of care homes, facilitating data collection of numerous patients in an environment that is less likely to alter their gait (*Robles-García et al., 2015*). Additionally, applications that do not require high accuracy will likely cope better with loose clothing.

### Diversity of human shapes and movements

While pose estimation algorithms are good at identifying keypoints from images they have been trained on, they can be poor at generalizing to identify keypoints in images that differ substantially from the training dataset (*Cronin, 2021*; *Mathis et al., 2020b*; *Seethapathi et al., 2019*). Image databases (*Chen, Tian & He, 2020*; *Ionescu et al., 2014*; *Lin et al., 2014*; *Sigal, Balan & Black, 2010*) may be biased towards humans of a certain race or a specific type of movement, and therefore, pose estimation algorithm performance may decrease when movements and people do not have sufficient representation (*e.g.*,

gymnastic movements (*Seethapathi et al., 2019*)). Manually labelled training datasets need to be diverse to account for varied movements of daily life (*e.g.*, walking, standing from a chair, picking up objects), sporting movements (*e.g.*, figure skating, gymnastics and weightlifting) and clinical movements (*e.g.*, neurological disorders and amputations), visual differences of participants (*e.g.*, age, race, anthropometrics) and visual differences of markerless setups (*e.g.*, lighting levels, scale of participant, camera angle). Because current pose estimation algorithms are trained to label each image in a video independently, they may perform well at detecting keypoints of patients with pathological gait abnormalities such as cerebral palsy and stroke, while physical abnormalities such as amputations will likely present a more difficult challenge. Clinical datasets could be collectively sourced from clinical research studies worldwide, however as standard video will be used to collect data, challenges in the form of patient confidentiality and ethical considerations must be overcome at the ethical application stage to achieve this.

## Shortcomings of current training datasets

Currently available open-source training datasets were never designed with biomechanical applications in mind. While these datasets encompass millions of images and thousands of manually labelled poses (*Lin et al., 2014*, *Andriluka et al., 2014*), only a subset of major joint centers have been labelled (ankle, knee, hip, shoulder, etc.), which increases errors as major joints are treated as a rigid segment (*Zelik & Honert, 2018*). For example, when walking with a fixed ankle/toe orthosis, markerless ankle joint angle (OpenPose) differences compared to marker-based methods were reduced, relative to normal walking, as toe flexion was not accounted for in normal walking by the markerless algorithm (*Takeda, Yamada & Onodera, 2020*). Additionally, open-source pose estimation algorithms that only detect joint centers struggle to identify more than 5DoF, as detecting rotation about the longitudinal axis requires three points on a segment.

Open-source manually labelled pose estimation training datasets (*Andriluka et al., 2014*; *Chen, Tian & He, 2020*; *Lin et al., 2014*) have recruited labelers from the general population who likely do not possess anatomical knowledge. As such, these datasets have not been labelled with the accuracy required for biomechanical applications, leading to errors in joint center locations and angles (*Needham et al., 2021b*). Furthermore, joints such as the hip or shoulder may appear very different from the side compared to a frontal or 45° angle. Evidence of this can be seen in the systematic offset of joint center locations and segment lengths outlined by *Needham et al. (2021b)*. Furthermore, open-source labelled datasets generally do not require all images to pass a second verification step, therefore two people may have very different interpretations of a joint center, which may lead to inconsistency in the labelled images (*Cronin, 2021*). It is unwise to expect pose estimation algorithms to match marker-based methods when the labelled data they are trained on is fundamentally flawed. Several commercial companies have created their own propriety datasets (*Kanko et al., 2021b*; *Kanko et al., 2021c*), with Theia3D employing trained labelers who likely have anatomical knowledge to label multiple points on each segment and have integrated a verification step by an expert labeler (*Kanko et al., 2021c*). This two-step labelling process

may produce a more biomechanically accurate dataset, enabling the strong results discussed previously (*Kanko et al., 2021a*; *Kanko et al., 2021b*; *Kanko et al., 2021c*).

Large open-source datasets have labelled keypoints even when joints are occluded. This is a requirement for entertainment applications as it would be unacceptable for limbs to suddenly go missing in video games or virtual reality. However, this results in occluded joints being labelled onto points that are biomechanically incorrect (*Lin et al., 2014*). For example, the right knee may be occluded by the left leg and thus labelled as being located somewhere on the left thigh. This results in two potential issues, firstly, the labeler must guess the location of the occluded joint, which reduces the accuracy of the dataset and secondly, the algorithm may learn that it is possible for joints to appear on locations that are biomechanically incorrect (*Cronin, 2021*). Finally, *Seethapathi et al. (2019)* highlighted that training and testing datasets often do not include temporal information (sequentially labelled images) and therefore current pose estimation algorithms can vary wildly in estimation of joint center locations between consecutive frames. However, it is possible to reduce these differences using Kalman filtering (*Needham et al., 2021a*) and as such, improvements to labelling of current open-source data sets (*e.g.*, COCO (*Lin et al., 2014*)) may be a more viable solution to improving accurate detection of joint center locations. New open-source datasets for biomechanical applications should include at least three points for each body segment, be labelled by trained labelers who possess anatomical and biomechanical knowledge, include a verification step by a secondary subset of expert users and additionally ignore or account for occluded joints.

## Evaluation

Current publicly available video datasets with synchronized marker-based motion capture, often use limited or sub-optimal marker placements, have low frame rates and camera resolution and thus may result in overestimating differences between markerless and marker-based systems compared to when run on private higher quality datasets (*Colyer et al., 2018*; *Corazza et al., 2009*). Publicly available evaluation data sets that include highspeed, high resolution images are needed for true comparisons between markerless and marker-based motion capture. While Needham and colleagues (*Needham et al., 2021b*) demonstrated that OpenPose had a greater difference to marker-based motion capture on average between 16 and 48 mm, maximal joint center location differences could be as high as 80 mm or even higher for some joints during running. Examining not only the accuracy, but the reliability of a system to accurately measure joint center locations is crucial, as systems are beginning to obtain average results that rival marker-based methods. However, we also need to question whether improving markerless motion capture methods to align closer to marker-based motion capture is the best solution. Marker-based motion capture has inherent errors discussed previously and markerless motion capture may potentially out-perform marker-based methods in some areas (*e.g.*, soft tissue artefact). As such, markerless methods next need to be assessed against bi-planar videoradiography or similarly accurate methods, to determine the true accuracy and reliability of these markerless systems.

## Decision making

Previous work has demonstrated the potential for markerless systems to automatically process video data and report quantitative results that could be immediately used by a clinician (*Kidziński et al., 2020*; *Martinez et al., 2018*). While pose estimation algorithms are learning to detect human poses, they are not able to think on their own. Desired outcome measures (*e.g.*, temporospatial measures and joint angles) extracted using pose estimation algorithms are still decided by humans. Emerging applications of markerless motion capture are therefore likely to require outcome measures to be chosen by the user prior to data collection, after which the markerless system will collect and process the data, similar to current implementations of commercial IMU systems (*i.e.*, Mobility Lab ADPM Inc.). As such, the clinician is still needed to interpret the results and their applicability to the patient. Deep learning methods could potentially be applied to this problem in the future (*Simon, 2004*), however, speculating on how this would be achieved is beyond the scope of this review.

## Usability

Current applications of open-source pose estimation algorithms require in-depth knowledge of deep learning-based neural networks and computer vision methods. As such, this technology requires usability improvements for users who do not have programming or computer science backgrounds. Some commercial systems such as Theia3D have made their software highly accessible by facilitating data collection using hardware and software of leading video-based motion capture companies (*e.g.*, Qualisys and Vicon). However, because they have a proprietary dataset and pose estimation algorithm, it is not possible for a user to determine what keypoints their algorithm is extracting.

While previous pose estimation algorithms have required substantial processing power housed in high end computers, new pose estimation algorithms can run on standard computers with modest graphical processing units (*Cao et al., 2018*) or even smaller devices such as mobile phones (*Bazarevsky et al., 2020*). As pose estimation software develops, it will become more feasible to integrate both the phone camera and processor to provide compact and affordable markerless motion capture (*Steinert et al., 2020*). Alternatively, cloud-based computing could be harnessed to record video using a smartphone, which is then uploaded to a server for processing, after which results are returned to the user (*Zhang et al., 2021*). Clinicians, researchers and coaches could one day perform automatic markerless motion capture in real time, without large setup costs. Finally, pose estimation algorithms have the potential to be used with cameras that move freely during data collection (*Elhayek et al., 2015*), which could allow accurate examination of how patients move through the natural environment.

## CONCLUSION

Markerless motion capture has the potential to perform movement analysis with decreased data collection and processing time compared to marker-based methods. Furthermore, markerless methods provide improved versatility of the data, enabling datasets to be re-analyzed using updated pose estimation algorithms and may even provide clinicians

with the capability to collect data while patients are wearing normal clothing. While markerless temporospatial measures generally appear to be equivalent to marker-based motion capture, joint center locations and joint angles are not yet sufficiently accurate. Current pose estimation algorithms appear to be approaching similar error rates of marker-based motion capture. However, without comparison to a gold standard, such as bi-planar videoradiography, the true accuracy of markerless systems is unknown. Current open-source pose estimation algorithms were never designed for biomechanical applications, therefore, datasets on which they have been trained are inconsistently and inaccurately labelled. Improvements to labelling of open-source training data will be a vital next step in the development of this technology.

### Funding
This work was funded by the EPSRC through CAMERA, the RCUK Centre for the Analysis of Motion, Entertainment Research and Applications, Bath, United Kingdom [EP/M023281/1 and EP/T014865/1]. The funders had no role in study design, data collection and analysis, decision to publish, or preparation of the manuscript.

### Grant Disclosures
The following grant information was disclosed by the authors:
EPSRC through CUK Centre for the Analysis of Motion, Entertainment Research and Applications, Bath, United Kingdom: EP/M023281/1, EP/T014865/1.

### Competing Interests
The authors declare there are no competing interests.

### Author Contributions
- Logan Wade conceived and designed this review, analyzed the data, prepared figures and/or tables, authored or reviewed drafts of the paper, and approved the final draft.
- Laurie Needham, Polly McGuigan and James Bilzon conceived and designed this review, authored or reviewed drafts of the paper, and approved the final draft.

### Data Availability
   This was a literature review and does not have raw data.

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
