# Peer review of "Applications and limitations of current markerless motion capture methods for clinical gait biomechanics"

_PeerJ, doi:10.7717/peerj.12995_

## Round 0.1 · original submission · Minor Revisions

This is a well written review. Please revise the manuscript based on the comments of the reviewers.

Reviewer 1 ·

Basic reporting

The manuscript is well written, clear, and unambiguous.

Experimental design

The manuscript presents a study well designed (scoping review).

Validity of the findings

Conclusions are well stated and linked to the original research question.

Additional comments

The manuscript presents a scoping review that discusses the applications and limitations of markerless motion capture. The authors should be commended for their insightful and well drafted manuscript. I only have a few minor comments that are mainly to help improve some sentences.

The only other point beyond typos is whether the authors should consider using the PRISMA extension for scoping reviews?
Tricco, AC, Lillie, E, Zarin, W, O'Brien, KK, Colquhoun, H, Levac, D, Moher, D, Peters, MD, Horsley, T, Weeks, L, Hempel, S et al. PRISMA extension for scoping reviews (PRISMA-ScR): checklist and explanation. Ann Intern Med. 2018,169(7):467-473



Abstract (line 28): should read ‘that are relevant’ rather than ‘that a relevant’.
Line 95: just for clarity, are these averages across various joints of the body/lower limb? Perhaps clarify for the reader.
Line 142-143: should this read ‘provided sufficient image resolution’ rather than ‘providing sufficient image resolution’?
Figure 4 caption: just a small type here with ‘estimation’ as ‘estimate4ion’.
Line 233: reword ‘outline’ to ‘outlined’.
Line 269: reword ‘as high has 20 cm’ to ‘as high as 20 cm’.
Line 415: remove the word ‘of’.
Line 514: here you mention IMU systems – I wonder if this should be mentioned earlier as another area of ‘markerless’ motion capture for clinical use? IMUs and other wearables are also increasingly being used for clinical use, and I know it is beyond the scope of this review, but may be worth a mention earlier on.

Reviewer 2 ·

Basic reporting

There are some areas that could be improved. Please see my additional comments.

Experimental design

N/A

Validity of the findings

no comment

Additional comments

The authors present an interesting and relevant review that is reasonably broad in scope. In general I enjoyed reading the work, but there are many cases where the wording is ambiguous and could be improved. This is important because the manuscript is currently difficult to read in places, which detracts from an otherwise interesting and timely paper. Specific suggestions are given below. Please keep in mind that I have been deliberately pedantic- these comments should not be interpreted as criticism, but rather as my attempt to help you improve what is already a solid piece of work, and one that is clearly worthy of publication.

Abstract, lines 20-1: “…this review explores how practical applications of this technology are being applied in clinical and rehabilitation settings” (double use of apply). I suggest re-wording to something like “this review explores how markerless approaches are being applied in clinical and rehabilitation settings”

27-29: “…and determine what gaps in the knowledge exist that a relevant to the
future directions and limitations of this developing technology”. I suggest simplifying: “and identify gaps in our knowledge that are relevant to future developments in this area”, or something similar.

29-30: “Markerless methods provide improved versatility of the data, enabling datasets to be re-analyzed using updated pose estimation algorithms”. How about “Markerless
methods increase data versatility, enabling datasets to be re-analyzed using updated pose estimation algorithms”.

31-4: “While it appears that markerless temporospatial measures generally appear to be equivalent to marker-based motion capture, joint center locations and joint angles are not yet sufficiently accurate. Current pose estimation algorithms appear to be approaching similar error rates of marker-based motion capture”. I think the flow here could be improved. For example, instead of the sentence starting “Current…”, could you instead state that it seems that error rates are improving quickly, resulting in progressively smaller errors between methods. You might also want to revise the double use of ‘appear’.

101-2: “an assumption that markers on the skin represent movement of the bone, leading to soft tissue artefact errors as muscle, fat and skin move beneath markers”. Could this be expressed more clearly? I think the assumption is that markers on the skin represent the position of the underlying bone. In addition, the problem is that the marker stays stationary on the skin, but that the skin moves relative to the underlying bone.

112: Markers-based should be marker-based

135: Reference instead of references

166-7: “…require increased lighting demands”. Suggest rephrasing

177 / Software: I leave it up to you, but it might be beneficial to introduce the concepts of supervised learning (first few paragraphs) and transfer learning (when introducing Deeplabcut) in this section.

201 (legend to fig 4): Remove ‘is’. Please also correct the spelling of ‘estimation’ in l203.

233: Outlined instead of outline

288-90: Could the authors expand on what they mean here? Specifically the ‘fundamental lack of data’ comment.

318: “Slower movements performed better” and “faster jumping and throwing movements were 20-40mm”. It’s obvious what is meant but I suggest revising this sentence- the movement itself cannot perform better.

330: Either remove ‘occurred’ or change to ‘occurring’.

330-2: I think this is a valid point, but why would the hip be the least accurate? If the problem were just poor labelling in general, then the errors for different body parts should be approximately the same. Could it be that the hip is more difficult to accurately identify just on the basis of an image (as opposed to physically palpating for the joint as you would in the lab), perhaps because it is (almost) always covered in images, whereas other body parts (e.g. ankle, wrist) may be easier to ‘see’ in an image?

339-40: “Another commercial system … recorded multiple movements … and then was processed”. Suggest revising.

363-75 / Practical applications: Here the different studies are introduced inconsistently. For some (Moro, Shin), the main findings are briefly summarised, whereas for others it is simply stated what was done. It would be good to use the former approach for all studies in my opinion- a brief sentence stating the key finding is sufficient, as done for the Moro/Shin studies.

393: Remove ‘be’

409: Please be more specific- greatly improve the efficiency of data collection? Or the ease?

415: “Therefore, markerless joint angle variability of may be similar to marker-based data collected on multiple days (inter-session). Remove ‘of’, but I also suggest revising. I think you mean that joint angle variability across multiple testing days may be similar between the two methods.

424: ‘this’ should presumably be ‘that’

483: ‘be’ instead of ‘are’?

496-8: “While Needham and colleagues (Needham et al. 2021b) demonstrated that OpenPose had a greater difference on average between 16-48 mm, joint center location differences could be as high as 80 mm or even higher for some joints during running”. This sentence lacks context. For example, ‘had a greater difference’ is not informative. I think you are referring to differences between OpenPose and a marker-based estimate, but this should be made clearer.

543-4: “While it appears that markerless temporospatial measures generally appear…”. Same comment as for the abstract- I suggest being a bit more assertive and avoiding the double use of ‘appear’, e.g. “While it appears that markerless temporospatial measures are generally equivalent…”

·

Basic reporting

This manuscript presents a scoping review of the markerless motion capture literature as it relates to clinical gait biomechanics. Overall, I find it to be a thorough, current, and balanced review of the state-of-the-art of markerless motion capture, which would be of interest for readers engaged with the topics of motion capture technology, clinical gait analysis, injury prevention and rehabilitation, and those pursuing clinical adoption of biomechanical measures.

The manuscript is written in clear and professional English and provides sufficient background/context and referencing of the literature. The review was clear with respect to its scope, and responsibly identified areas that were not within the scope but have been discussed elsewhere and directed readers to those works where appropriate. While the field of markerless motion capture has been reviewed previously, this scoping review provides an updated view of the literature from a clinical biomechanics perspective that will be accessible to a wide audience of biomechanics researchers.

Experimental design

The manuscript is within the aims and scope of the journal and meets the definition of a scoping review described in Munn et al. (2018). The description of the survey methodology includes the literature databases that were searched, the search terms that were used, and the publication date restrictions (none). This survey methodology meets the broader inclusion process of scoping reviews, and there were no apparent omissions of literature. Sources are cited and appropriately paraphrased. The structure of the article is logical, and the included figures effectively serve to illustrate the concepts discussed within the review.

Validity of the findings

This manuscript does not assess impact nor novelty. The methodology is sufficiently described to allow replication of this scoping review, and replication of existing works that may benefit the literature is encouraged through the discussion and conclusions (i.e. improving and expanding biomechanically-oriented image training datasets). Conclusions are appropriately drawn from the review of existing literature and identify areas for continued investigation and improvement by the field.

Additional comments

Overall, this scoping review provides a useful clarification of the state-of-the-art of markerless motion capture in the context of clinical biomechanics and effectively identifies gaps in the knowledge and literature that require further exploration. Conclusions are drawn from the reviewed literature and provide clear direction for future research.

Additional minor comments and suggested revisions have been added to the manuscript pdf and attached to this review.

---

## Round 0.2 · accepted · Accept

Please make the recommended minor changes from reviewer #3. Congratulation for the successful revision!

Reviewer 1 ·

Basic reporting

The manuscript is well written, clear, and unambiguous.

Experimental design

The manuscript presents a study well designed (scoping review).

Validity of the findings

Conclusions are well stated and linked to the original research question.

Additional comments

The authors have responded well to all comments from reviewers. I have no further comments to make - the authors should be commended for this interesting and timely scoping review.

Reviewer 2 ·

Basic reporting

no comment

Experimental design

no comment

Validity of the findings

no comment

Additional comments

I thanks the authors for their revision. In general I am satisfied that the clarity has improved and that the paper is suitable for publication.

·

Basic reporting

The revised manuscript improves upon the original submission with several corrections and clarifications to the language and grammar used. The writing is clear and unambiguous, and I have no issues with the reporting.

This review follows a logical structure and order, and the topic is adequately introduced. The scope of the article is clear and topics that are related but outside the scope are identified as such. The writing is consistent and should be understandable by the target audience.

Experimental design

The manuscript is rigorous and performed to a high technical and ethical standard, maintaining objectivity. The review covers all relevant topics and points readers to additional materials for further detail on related subjects where appropriate.

Validity of the findings

The conclusions and main future directions are well-founded, clear, and supported by referenced works and findings. Work towards closing the identified knowledge gaps and unresolved questions is encouraged and appropriate suggestions are provided for improvements in these areas.

Additional comments

Overall, I find this review to be well-written and provide a clear overview of the current state of markerless motion capture for clinical gait biomechanics. Besides the following very minor points, I have no suggested changes.

‘Pose’ is clarified as the combination of position and orientation in several places (lines 63, 117, 220) – only the first time is required.

Line 414: “…or may not” is not necessary, and missing apostrophe from “a pose estimation algorithm’s ability”.

Line 456: References to datasets “[55-58, 102]” appear to be in a difference style.